# The Genetic Basis of Phosphorus Utilization Efficiency in Plants Provide New Insight into Woody Perennial Plants Improvement

**DOI:** 10.3390/ijms23042353

**Published:** 2022-02-21

**Authors:** Yanjun Pan, Yuepeng Song, Lei Zhao, Panfei Chen, Chenhao Bu, Peng Liu, Deqiang Zhang

**Affiliations:** 1National Engineering Laboratory for Tree Breeding, College of Biological Sciences and Technology, Beijing Forestry University, No. 35, Qinghua East Road, Beijing 100083, China; yanjunpan@bjfu.edu.cn (Y.P.); yuepengsong@bjfu.edu.cn (Y.S.); zhaolei@bjfu.edu.cn (L.Z.); panfeichen@bjfu.edu.cn (P.C.); buchenhao@bjfu.edu.cn (C.B.); ackliup@163.com (P.L.); 2Key Laboratory of Genetics and Breeding in Forest Trees and Ornamental Plants, College of Biological Sciences and Technology, Beijing Forestry University, No. 35, Qinghua East Road, Beijing 100083, China

**Keywords:** phosphate starvation response, signal transduction, transcription factor, microRNA, phosphate transporter

## Abstract

Soil nutrient restrictions are the main environmental conditions limiting plant growth, development, yield, and quality. Phosphorus (P), an essential macronutrient, is one of the most significant factors that vastly restrains the growth and development of plants. Although the total P is rich in soil, its bio-available concentration is still unable to meet the requirements of plants. To maintain P homeostasis, plants have developed lots of intricate responsive and acclimatory mechanisms at different levels, which contribute to administering the acquisition of inorganic phosphate (Pi), translocation, remobilization, and recycling of Pi. In recent years, significant advances have been made in the exploration of the utilization of P in annual plants, while the research progress in woody perennial plants is still vague. In the meanwhile, compared to annual plants, relevant reviews about P utilization in woody perennial plants are scarce. Therefore, based on the importance of P in the growth and development of plants, we briefly reviewed the latest advances on the genetic and molecular mechanisms of plants to uphold P homeostasis, P sensing, and signaling, ion transporting and metabolic regulation, and proposed the possible sustainable management strategies to fasten the P cycle in modern agriculture and new directions for future studies.

## 1. Introduction

Plant growth and development rely on the availability of appropriate composition and concentration of nutrients in the soil. Insufficient, limited, unbalanced, or excessive supply of these nutrients in the soil system confine plant growth, development, yield, quality, and fertility. Phosphorus (P), a critical macronutrient, incredibly affects the growth and development of plants. Not only does it play a structural part in myriad biologically important molecules such as deoxyribonucleic acid (DNA), ribonucleic acid (RNA), adenosine triphosphate (ATP), nuclear protein, phospholipids, sugars, and phosphorylated metabolites, but also it acts as a vital substrate for a host of physiological processes, including photosynthesis, respiration, signal transduction, and energy metabolism [1]. Moreover, the activities of several vital enzymes involved in C and N metabolism are decreased by P starvation [2,3]. Pi is the inorganic form of P absorbed and metabolized by plants as well as a pivotal substrate of most biochemical reactions in cells [4,5]. Though P is one of the most critical macronutrients on the Earth, it is mostly unavailable to plants because of the conversion of Pi into organic compounds by microorganisms, the fixation with metals, and the low diffusion rate [6]. Therefore, Pi is usually a restricting agent in agro- and forest-ecosystems. Unfortunately, Pi rock is a finite and unrenewable resource serving as the major source of P fertilizers, which is estimated to be exhausted in the next 50 to 100 years [7]. Apart from the decreasing Pi rock reserves, the declining quality, the uneven geopolitical distribution, and the growing costs have all deteriorated people’s concerns about improving the utilization rate of P [8,9,10]. How to realize the sustainable utilization of Pi in the process of plant cultivation has become the main goal of plant breeding. In this review, we summarized recent progress in mechanisms underlying P availability-responsive P acquisition, effects of environmental factors on P acquisition, the transportation, storage, and metabolism of P, and the genetic basis of P utilization.

## 2. Signal Transduction Pathway of P Environmental Perception

Both Pi starvation and excessive Pi have harmful effects on plants. On the one hand, a great deal of biosynthesis of macromolecules as well as many biological processes could be impaired in plants due to Pi starvation [11,12]. On the other hand, serious cellular damage and growth retardation could be caused by excessive Pi in cytoplasms [13,14,15,16]. Therefore, it is of great essentiality to quickly perceive the Pi levels in vivo and in vitro, then to take appropriate measures to maintain Pi homeostasis in living organisms. So far, immense headway has been made on the mechanism of how the Pi level is perceived by single-cell organisms, however, little is known about it in multi-cellular eukaryotes including plants and animals [17,18,19]. It is the root tips that sense Pi status, which has been approved by a fact that *Arabidopsis*’ impaired root tips could not absorb Pi from the medium in 10 min [20,21]. Firstly, roots sense fluctuations in extracellular nutrient levels and send signals to shoots through the xylem, as a warning that a particular nutrient supply is about to be limited [22,23,24]. Subsequently, these nutrient signals from roots are perceived by shoots and some signals are transmitted to stem tips and roots through the phloem to manage plant development and nutrient uptake [24].

Genes that participated in Pi absorption and recovery and lipid metabolism are in a general way modulated systemically, while genes tied into stress- or hormone-related reactions are regulated locally [17,25]. Such regulation should work sequentially and in a specific manner, which requires sophisticated communication and local and systematic signal transmissions [17]. Sensing and signaling systems, which report alterations in internal and external states and trigger corresponding responses, govern plant growth by integrating endogenous signals and environmental agents [17]. It is the fact that the growth of shoots and roots rely on each other which underlines the significance of integrating signaling pathways [17].

Inositol polyphosphates (InsPs), highly phosphorylated molecules, are diverse Pi storage and signaling molecules pervading in plants and animals and are implicated in numerous biological processes including hormonal signaling, energy sensing, and responses to biotic and abiotic stresses [26,27,28,29]. Moreover, InsPs might be further phosphorylated on existing Pi groups to generate inositol pyrophosphates (PP-InsPs) which contain one or more high-energy pyrophosphate groups [30]. Recent studies have regarded SPX domains as high-affinity PP-InsP sensors in eukaryotes [31]. Evidence suggests that SPX-containing proteins are closely related to Pi sensing and metabolism in eukaryotes [30,32].

For example, SPX1 and SPX2 in *Arabidopsis* and rice, SPX4 in rice work as repressors of the Pi signaling pathway to prevent the nuclear localization of PHOSPHATE STARVATION RESPONSE 1/PHOSPHATE STARVATION RESPONSE 2 (AtPHR1/OsPHR2) in a Pi dependent manner [15,33,34,35]. SPX domains own a highly basic binding surface that has a strong affinity with PP-Insps [31], displaying that PP-Insps are included in the regulation of Pi signaling mediated by SPX [30]. Up to date, there are still an array of sensors that are ambiguous and need to be investigated.

## 3. P Absorption, Transportation, Storage, and Metabolism

A host of Pi transport systems are required to tune up the acquisition of Pi from varying environments and its ensuing transport within cells of plants [6,36]. In fact, a dual mechanism aiming at disparate Pi environments is derived in plants, namely a high-affinity system and a low-affinity system [37,38]. P low-affinity transporters are generally found at high concentration levels (mM), while P high-affinity transporters are mainly detected at low concentration levels (μM) [39,40].

Plants actively absorb Pi in the soil mainly through the reverse concentration of the root hair zone. To be more specific, it is generally believed that the active P absorption process is driven by the H^+^ of H^+^-ATPase on the vacuole membrane. And the proton phosphoric carrier, the mode of H^+^ co-transport with H_2_PO_4_, realizes the process [38,41,42].

After Pi enters the cell, Pi is then distributed to sub-organelles by Pi transporters to retain the cellular normal metabolic activity [11]. For instance, Pi could be transported between chloroplasts/plastids and mitochondria, where it is absorbed by photooxidation and oxidative phosphorylation [43]. Furthermore, Pi entering the root system is shipped to the surface by PHOSPHATE1 (PHO1) located in the root vascular tissue, thereby forming a system for transportation from the root system to the surface [11]. P has obvious regionalization in cells and plant tissues. Cells and tissues are separated into contrasting regions because of the complex membrane systems in plants. Generally speaking, most of the Pi is found in the vacuole and only a small part exists in the cytoplasm and organelles. Vacuole acts as the reservoir of cellular P, while the cytoplasm is the metabolic pool of cells [44]. The destination of P after entering the plants is determined by the needs of divergent tissues and metabolic processes, mainly including: (1) participating in the synthesis of phospholipids, DNA, RNA, and other living substances; (2) transporting to plastids or mitochondria to participate in the metabolic processes; (3) storing in vacuoles; (4) transferring to other cells [45,46].

## 4. The genetic Basis of P Utilization

To sustain Pi homeostasis, plants have promoted diverse adaptive responses at the transcriptional, post-transcriptional, and post-translational levels, collectively known as the Phosphate Starvation Responses (PSRs) [40,47]. Plentiful transcription factors (TFs) and transporters are allied to the Pi signaling pathway, which are regulated by sugar, phytohormone, and photosynthesis signaling [47,48].

### 4.1. Transcription Factor Regulating P Homeostasis

Several vital TFs in the Pi signaling pathway have been identified and functionally characterized in plants, including members of the MYB, WRKY, bHLH, ZAT families, etc., which could conduct Pi homeostasis and signaling in plants positively or negatively [49,50,51,52]. Thereinto, TFs involved in Pi signaling are principally listed in Table 1.

Lots of genes encoding phosphate starvation-induced (PSI) protein phosphate transporters (PT) are transcriptionally induced by Pi deprivation, some of which are under the control of central governors of Pi starvation signaling, the phosphate starvation response (PHR) TFs [49,54,63]. PHOSPHATE STARVATION RESPONSE 1 (PHR1) and PHR1-LIKE1 (PHL1)—which orthologs have been recognized in offbeat species—play a focal role in the regulation of PSRs in *Arabidopsis* [49]. In *phr1* and *phr1 phl1* mutants, there are some transcriptional activation and repression changes to Pi starvation [17,54]. *PHR1* exerts a great influence on regulating genes touched upon Pi signaling, anthocyanin biosynthesis, carbohydrate metabolism, whereas its role is thought insignificant in membrane lipid remodeling [49,63,64,65]. In addition, an MYB transcriptional activator of *INDUCED BY PHOSPHATE STARVATION 1* (*AtIPS1*) and *RIBONUCLEASE 1* (*AtRNS1*), the AtPHR1 TF, could be sumoylated by AtSIZ1, a small ubiquitin-like modifier (SUMO) E3 ligase [66]. *PHR1* and *PHL1* could modulate the transcription of its downstream genes by combining the cis-acting element PHR1-binding site (P1BS) or P1BS-like elements, a DNA motif of GNATATNC or AC/AATATT/CC, respectively, on the promoter of target genes during the Pi starvation stress, such as several *PHOSPHATE TRANSPORTER 1* (*PHT1*) genes, *PHOSPHATE TRANSPORTER TRAFFIC FACILITATOR 1* (*PHF1*) genes, MIR399 and MIR827 [12,49,53,67]. OsPHR2 TF, the PHR1 ortholog in rice, responds to variations in Pi status via altering its subcellular localization, which is isolated in the cytosol when rice is exposed to high Pi conditions and shifts to the nucleus when Pi is scarce [35,68].

Besides *PHR1* and *PHL1*, there are some other MYB transcription factors that are related to PSR, such as *MYB62* and *MYB2* [56]. It was found that overexpression of *MYB62* could suppress the expression of a large quantity of PSI genes, gibberellic acid (GA) biosynthetic genes and affect Pi uptake, and acid phosphatase activity [17,56]. The expression of *miR399f* is upregulated by *MYB2* in *Arabidopsis* due to the combination of MYB-binding sites (MBS; TAACTG) in its promoter region, subsequently, the expression of several PSI genes is increased [57]. In summary, *AtMYB62* is a transcriptional suppressor, while *MYB2* activates target gene’ transcription during stress [56,57].

WRKY transcription factors, which function as positive or negative regulators, are involved in PSRs in *Arabidopsis* such as *AtWRKY6*, *AtWRKY42*, *AtWRKY45,* and *AtWRKY75* [58,60,62]. *AtWRKY6* with a C2H2 zinc finger domain exerts negative regulatory effects on *PHO1* expression in a Pi-dependent manner via binding to W-box (TTGACT/C) motifs located on the *PHO1* promoter [17,69,70]. Furthermore, *AtWRKY6* has been proved to be positively implicated in the regulation of response to boron (B) deficiency [71,72]. *AtWRKY42*, a homolog of *AtWRKY6*, negatively modulates *PHO1* transcription and positively conducts *PHT1* expression [60]. They work redundantly in *Arabidopsis* Pi translocation through down-modulating *PHO1* expression, while they have nonredundant functions in Pi acquisition because *PHOSPHATE TRANSPORTER 1;1* (*PHT1;1*) expression is activated by *AtWRKY42* under Pi-sufficient conditions and is repressed by *WRKY6* under Arsenate stress [73]. Both WRKY6 and WRKY42 are degraded via 26S proteasome-mediated proteolysis in the PSR [58,60]. *AtWRKY45* and *AtWRKY75* are upregulated in Pi-deficient conditions [62,74]. RNAi knockdown studies manifested that a reduction in *WRKY75* and *WRKY45* mRNA levels reduced the up-regulation of several PSI genes, shrinking Pi absorption, increased anthocyanin accumulation, and enhanced growth of lateral roots and root hairs [62]. Recently, studies have uncovered that *OsWRKY21* and *OsWRKY108* act redundantly to promote Pi uptake in rice via activating *OsPHT1:1* expression under Pi-replete conditions [52]. *OsWRKY74* also modulates tolerance to Pi starvation in rice [75]. Interestingly, *OsWRKY74* might also be in connection with the response to deficiencies in iron (Fe) and nitrogen (N) as well as cold stress [75]. Compared to WT plants, *OsWRKY74*-overexpressing plants possessed a higher accumulation of Fe and up-regulation of the cold-responsive genes under the Pi deficiency condition [75].

Basic helix-loop-helix (bHLH) TFs have been characterized so far. For example, basic helix-loop-helix 32 (*AtbHLH32*) in *Arabidopsis* and *OsPTF1* (Pi Starvation-induced Transcription Factor 1) in rice are upregulated under Pi starvation [50,76]. *AtbHLH32* and *OsPTF1* play completely opposite roles in PSR respectively, of which *AtbHLH32* acts as a negative regulator of PSR [50,76]. Compared with the wild-type plants, the anthocyanin accumulation, PSI gene expression and total Pi content of the *bhlh32* mutant were significantly increased [50,70]. The function of TTG1-containing complexes is interfered with by bHLH32 and a wide range of biochemical and morphological processes responding to Pi availability are influenced [50]. *OsPTF1*-overexpressing rice indicated that an enhanced tolerance to Pi starvation might be due to the changed root architecture [76].

*ZAT6* (*ZINC FINGER OF ARABIDOPSIS THALIANA 6*), a cysteine-2/histidine-2 zinc finger transcription factor, is involved in modulating Pi homeostasis and Root system architecture (RSA) [61]. Pi deficiency could induce the expression of *ZAT6* which functions as a repressor. The expression levels of some PSI genes are inhibited in overexpressing-*ZAT6* plants [61]. In addition, overexpressing-*ZAT6* plants show a decrease in seedling growth and changes in RSA, which have nothing to do with the state of Pi [61]. *AUXIN RESPONSE FACTOR 16* (*OsARF16*) and *AUXIN RESPONSE FACTOR 12* (*OsARF12*), two rice transcription factors in the ARF gene family, are implicated in PSR in rice as well [77].

### 4.2. Phosphate Transporters

Phosphate transporters could further be regulated at the post-translational level either via the abundance of protein or the subcellular localization to keep Pi homeostasis. The process of Pi transport from the soil solution into the root cells and subsequently the distribution of Pi in diverse parts of plants require the participation of functionally specific phosphate transporters, which are classified as follows [4,17,51].

Proteins that contain the SPX domain are divided into the following four families relying on the presence of additional protein domains: SPX, SPX-MFS, SPX-EXS, and SPX-RING [17,32]. Studies have shown that members of the SPX-MFS family are located in the vacuole membrane and participate in Pi transport [78]. In rice, the SPX-MFS family containing OsSPX-MFS1, OsSPX-MFS2, and OsSPX-MFS are all located on the vacuole membrane [79]. Among them, OsSPX-MFS3 is involved in the inflow of Pi from the vacuole into the cytoplasm, whereas OsSPX-MFS1 may mediate the inflow of Pi from the cytoplasm into the vacuole [44,80]. AtPHO1 functions in the Pi loading into the xylem and transporting to shoot [81,82]. Though the Pi export could not be regulated by the EXS domain, the EXS domain of PHO1 is vital for the Pi transportation and localization to the Golgi and trans-Golgi network [83,84]. The SPX-RING family has a zinc finger domain which is related to the interactions between proteins [84]. Some studies in *Arabidopsis* have suggested that the interactions between proteins—which is the SPX domain—are implicated in the Pi transposal [85,86]. NITROGEN LIMITATION ADAPTATION (NLA), a member of the SPX-RING family, is associated with the regulation of Pi homeostasis in plants and the regulation depends on N [87,88]. Notably, the coordinated utilization of N and P is crucial for maintaining optimal plant growth and achieving maximal crop yield [68]. NITRATE TRANSPORTER 1.1 B (OsNRT1.1B)-OsSPX4-OsPHR2 signaling module is responsible for the transduction of nitrate signal to activate PSRs [68].

Based on functional differences and subcellular localization, PHT transporters in plants are roughly classified into four families: PHT1, PHT2, PHT3, and PHT4, which have a distinct number of genes in various plants [44,46,89,90]. There are 12, two, nine, and six members of PHT1, PHT2, PHT3, and PHT4 families in poplars, respectively [91]. Most of the members of the four families are respectively located in the plasma membrane, chloroplast, mitochondria, and diverse subcellular compartments such as the Golgi apparatus [5,43,89,91]. The PHT1 family which mainly serves as H+/Pi symporters implicated in the absorption of Pi from various soil and transportation within plants [5,92]. PHT1 transporters of diverse species are mainly located on the plasma membrane (PM) of root cells, in accordance with their translocation of Pi from the apoplast into the cytosol [58,93,94,95].

The C-terminal of PHT1 has a few phosphorylation sites, and these sites are joint to the post-transcriptional control of the protein in *Arabidopsis thaliana*, emphasizing the importance of post-translational modification (PTM) in the integration and regulation of Pi metabolism in plants [96,97]. Compared with the PHT1 family, there are few studies on the other three PHT families, which are mainly responsible for Pi transport between organelles [98]. Photosynthesis and respiration are collectively ensured by PHT2, PHT3, and PHT4 families’ transporters.

In addition to the SPX domain-containing phosphate transporter and PHT transporters, there are a vast array of transporters taking part in the regulation of P homeostasis. For example, triose-phosphate translocator (TPT) is located in the membrane of the chloroplast envelope in spinach, potato, pea, maize, where it controls the counter exchange of triose phosphates or 3-phosphoglycerate and Pi between the stroma of the chloroplast and the cytosol of the cell [99]. Mitochondrial phosphate transporter (MPT) is responsible for transporting Pi into the mitochondrial matrix, where Pi acts as a reactant for the conversion of ADP to ATP [100,101]. The processes of Pi^−^/H^+^ symport or Pi^−^/OH^−^ antiport and Pi^−^/Pi^−^ exchange are mediated by MPT [100,101]. Up to now, a large number of transporters associated with P transportation still have not yet been identified, so it is of great necessity for us to do more work.

### 4.3. Phosphate Signal Transduction Pathway

Plant growth is regulated by the integration of endogenous signals and environmental factors through sensing and signaling systems which could report alterations in internal and external states and trigger corresponding responses to these alterations [17]. Signaling is divided into two types roughly. One is a local signaling that relies on external Pi concentration, the other is a systematic or long-distance signaling depending on the Pi state of the whole plant [17]. The former is responsible for initiating adjustments in RSA to enhance Pi acquisition, while the systemic, or the latter serve to modulate Pi uptake, mobilization, and redistribution [20,102,103,104]. A host of signaling molecules are derived when sensing the alterations of external Pi concentrations such as sugars, miRNA, and hormones. Moreover, Pi signaling and other signaling pathways which are composed of these signaling molecules are cross regulated [105].

Root sense fluctuations in extracellular Pi levels and send signals to shoots through the xylem, then these signals derived from the roots are sensed by the shoots and sent to the stem tip and roots via the phloem to regulate the development of plants and Pi absorption [22,23,24,70]. An overall hypothetical mechanism that illustrates the Pi sensing and signaling has been proposed in plants in Figure 1. Pi availability is first perceived by roots that are capable of monitoring the Pi concentrations in soils, and there appear to be two ways for plants to sense exogenous Pi. One is that the exogenous Pi is transported into the cell through the plasma membrane, thus being sensed in the cell; and the other is that the external Pi is sensed by sensors or receptors that are not yet recognized in the plasma membrane via decoding the external Pi conditions into internal signals [17,104,106]. Some studies have suggested that the triggers of PSR come from inside rather than the external Pi environment [107,108]. Due to the existence of NRT1.1 nitrate transporter as a nitrate sensor in *A. thaliana* and the PHO84 transporter that senses and transports Pi in yeast, we need to perform more studies to identify the processes of the external perception of Pi in plants [109,110]. With regard to the perception of the internal Pi, some people have presented that it is sensed by internal sensors at the root tip, although that has not been proven up to now. The question of whether Pi availability is sensed externally, intracellularly, or through a combination therein is worthy of further exploration. The input of primary signals starts the Pi signaling pathway and are converted into secondary signal components including hormones, sugars, Reactive Oxygen Species (ROS), Ca^2+^, and miR399 through the primary module, which is a critical step in amplifying a single input into distinctive responses. The secondary signals are interconnected and mutually modulated to regulate the development of final responses. In addition, communication between the two modules and secondary signal molecules cannot be circumvented, allowing for the use parallel pathways and positive or negative feedback loops to fine-tune the magnitude of output [17].

To deal with Pi starvation, plants have evolved a series of mechanisms to optimize Pi acquisition from the soil as well as its distribution to distinct organs and sub-cellular compartments [4,40], including the alterations of RSA such as the increase of root hairs, induction, or enhancement of the expression of high-affinity Pi transporters and the rise of root exudates such as phosphatases, having certain promoting effects on the acquisition of P in the root system. These adaptive PSRs work together by integrating and coordinating distinct signaling pathways, which are shown in Figure 2. As a matter of fact, there are not only transcriptional events but also posttranslational events in plants, which are generated to modulate a series of PSRs and then to hold the Pi homeostasis under Pi deficiency conditions [40]. The rough processes are shown in Figure 2 and Figure 3.

## 5. Epigenetic Basis of P Utilization

A great many non-coding RNAs—including MicroRNAs (miRNAs) and long non-coding RNAs (lncRNAs)—are related to the regulation of P homeostasis [111,112]. Up to now, the role of miR399 in the regulation of plant Pi homeostasis has been identified [113,114]. It was reported that miR399 is highly upregulated under Pi-deprivation conditions, and subsequently, the transcript levels of its target gene, a recognized ubiquitin-conjugating E2 enzyme (UBC24) which has five complementary sequences to miR399 in its 5′UTR, is restrained [111,115,116]. Furthermore, on the basis of a molecular genetic study of *pho2* mutant, *PHOSPHATE 2* (*PHO2*) was determined to be the locus of *UBC24* [12,117]. It guides the cleavage of *PHO2* mRNA which encodes UBC24 [17]. Overexpressing miR399 shows the overaccumulation of Pi in shoot and leaves that are caused by the increase in Pi absorption and translocation from the root to the shoot and the decreased remobilization from mature leaves to younger tissues respectively. The phenotype is consistent with the *pho2* mutant and the *ubc*24 knockout lines [12,117,118,119].

Induction of miR399 under Pi starvation is positively regulated by PHR1 and by the availability of photosynthetic products, while its activity on the cleavage of *PHO2* mRNA is restrained by *A. thaliana 4/INDUCED BY PI STARVATION1* (*At4/IPS1*) RNAs [49,53,120,121]. *At4* and *IPS1*, and their orthologues, *M. truncatula 4* (*Mt4*) and *TOMATOPHOSPHATE STARVATION-INDUCED GENE 1* (*TPS1*), are highly Pi-starvation-induced lncRNAs [67,122]. *AT4/IPS1* RNAs play the role of riboregulators and interfere with miR399 targeting of *PHO2* mRNA, which is called target mimicry [49,53,120,121]. MiR399 could directly cut the *PHO2* gene in some plants, which is affected by the long-chain non-coding RNA *IPS1/2*. *IPS1/2* could inhibit miR399 targeting *PHO2* by mismatching with miR399 [117,122]. Compared with wild-type plants, under the stress of P deficiency, the *at4* mutants showed the defects of Pi redistribution between shoots and roots, and excessive accumulation of Pi existed in shoots, while the overexpression of *IPS1* led to the decrease of Pi level in shoots, indicating that these *At4/ IPS1* family genes played an important part in P homeostasis [122]. MiR399 is conserved across plant species and regulates phosphate homeostasis by regulating the ubiquitin-conjugating E2 enzyme PHO2 that regulates PHO1 and PHT1 phosphate transporters [123]. In summary, miRNA399 takes part in the P deficiency signaling pathway from shoot to root [120]. Through phloem movement and inhibition of E2 binding enzyme, it leads to an increase in the expression of P uptake transporter in roots, which promotes P absorption by roots, transports and distributes it to the shoot [120]. The miR399-mediated PHO2 regulation is not only related to Pi homeostasis but also helps to control the development of stomata [123]. The stomatal lineage miRNA is a critical component of the regulation mechanism of stomatal development [123].

Besides miR399, plenty of miRNAs are found to take part in the regulation of P-homeostasis, such as miR169, miR395, miR398, miR827, and miR828. Both *Arabidopsis* and rice miR827 are induced by Pi starvation, the difference is that miR827 in rice targets two particular proteins, including SPX and MFS, while *Arabidopsis* miR827 targets *NLA* that encodes a protein consisting of an N-terminal SPX domain and a C-terminal RING domain, which is associated with N-homeostasis [86,113,114,124]. No matter under Pi deficiency or N deficiency conditions, the expression of miR169 is down-regulated. On the contrary, sulfur and copper deficiency could respectively induce the expression of miR395 and miR398 which are inhibited under Pi deficiency. [113,114]. In addition, it was also showed that PURPLE ACID PHOSPHATASE 1/MYB DOMAIN PROTEIN 75 (PAP1/MYB75) modulates anthocyanin biosynthesis during Pi deficiency through the autoregulation mechanism of miR828 and Tas4-siR81 (−), a trans-acting siRNA serving as a regulator about the biosynthesis of anthocyanin under Pi deficiency condition [17,113]. To sum up, non-coding RNA plays a vital role in regulating PSR.

## 6. Research Progress on Utilization of P in Forest Trees

Recently, some advances have been made in the utilization of P in trees and other perennials. *Eucalyptus grandis* is one of the most extensive tree species planted all over the world and is confronted with serious drought on account of the not fully developed root system in the first months [125]. Researchers took a complete randomized approach to study the influence on *E. grandis* such as morphological traits, architectural traits, and physiological traits under different conditions, involving sufficient water and P application conditions, sufficient water and no P application conditions, water stress and no P application conditions, and water stress and P application conditions [125]. *E. grandis* seedlings grown under good watering and P application conditions have better growth and higher photosynthesis indicators mostly as a result of the improvement of plant N nutrition, which suggests that the demand for P in the early growth stage of *E. grandis* is higher; however, under drought stress, P application may only affect the physiological and biochemical indicators of seedlings such as the increase of leaf relative water content (LRWC), net photosynthetic rate (Pn), optimal/maximal quantum yield of PSII (Fv/Fm), chlorophyll, nitrogen-containing compounds in seedlings, and the decrease of peroxidation of membrane lipid, but it has no effect on the morphological characters of seedlings [125]. Taking these together, P could help to deal with future climate change such as drought. Similar phenomena are also found in *Phoebe zhennan* (Gold Phoebe). In addition, the contents of hydrogen peroxide (H_2_O_2_) and malondialdehyde (MDA) and the activities of superoxide dismutase (SOD), peroxidase (POD), catalase (CAT), acid phosphatase (ACP), and purple acid phosphatase (PAP) were influenced by different P stresses in apple seedlings [126].

Researchers used a similar design to investigate the effects on Gold Phoebe, getting similar results. Moreover, it has been found that plant drought resistance could be enhanced by root system interaction with *Arbuscular mycorrhiza* (AM) since the ABA-mediated abiotic signaling pathways involving D-myo-inositol-3-phosphate synthase (IPS) and 14-3-3 proteins are partly regulated by fungi [127]. The improvement of the root system to absorb soil moisture is caused by higher root biomass [127]. To this end, the drought resistance of Gold Phoebe might be enhanced by improving the P efficiency in soil [128]. Scientists also found similar results in *Alnus Cremastogyne*, a fast-growing broad-leaved tree species unique to southwestern China, that is the application of P fertilizer, or the advancement of P use efficiency could improve the resistance to respond to drought according to lots of evidence, involving the increased LRWC and Pn [129,130,131]. *A. cremastogyne*—which has not only imperative commercial value but also restoration value—is a vital tree species on the earth [131,132]. The enhancement and facilitation of metabolism in *A. cremastogyne* are mainly caused by biochemical and physiological changes instead of morphological changes [131]. In other words, the negative influence of drought may be eased by P fertilizer or enhanced P use efficiency through modulating the antioxidant and osmotic potential [131]. *Machilus pauhoi* is an excellent evergreen broad-leaved tree species in the subtropics of China, mainly distributed in Guangdong, Guangxi, Jiangxi, Fujian, and Zhejiang, and other places, which likes shade environment in juvenile and prefers to light and humidity environment in adults. For example, the biomass relative growth rate of the whole plant displayed a significant difference in response to N and P application in different seasons: the biomass relative growth rate in summer is greater than that in autumn, while in autumn, it is greater than that in spring, and in spring, it is greater than that in winter [51]. In addition, each season showed a trend of increasing first and then diminishing [51]. Interestingly, approximately 75% of inorganic and 60% of organic foliar Pi detected in May were remobilized by November in *Populus alba* (white poplar) [133]. P is remobilized from senescing leaves in autumn and stored in other tissues for reuse in the following spring in deciduous trees [133].

Roots of young *F*. *sylvatica* L. trees in their native soils from two forests sites with strikingly different availability of P: one P rich and P poor soil were studied [134]. It has been uncovered that alterations in the fine root proteome of *F*. *sylvatica* L. trees are linked with P-deficiency and amelioration of P-deficiency [134]. Next, a similar study to inquire how long-lived forest trees deal with low soil P availabilities was conducted, which is concerned with P nutrition of beech in soils from P-rich and P-poor beech forests throughout an annual growth cycle [135]. In this study, P absorption, P content, and biomass were analyzed during five phenological stages including dormancy in winter, bud swelling in early spring, mature leaves in early and late summer, and senescent leaves in autumn [135]. Seasonal distribution patterns indicated that young leaves and emerging leaves were preferred sinks for P under P-poor conditions, thereby maintaining the leaf P concentration at levels similar to those of trees grown in P-rich soil [135]. No matter in low or high P conditions, coarse roots are the central P storage tissue that supply Pi to newly formed leaves [135]. There are different phenomena in P-poor soil and P-rich soil. On the one hand, beech trees in P-poor soil showed an increase in net biomass in the early annual growth period, accompanied by a strong P deficiency, which was replenished by strengthened absorption in late summer and autumn [135]. On the other hand, trees in P-rich soil grew until late summer, and P in the organic pools decreased moderately and recovered in the late fall, which coincided with the increase in P uptake from soil [135]. Recently, large-scale studies have detected that carbohydrate depletion brings about decreased P absorption, heightened internal P remobilization, and root biomass trade-off to compensate for the P squeeze [136]. Similar to herbaceous plants, trees are more prone to drought due to reduced root biomass [136]. The results associated tree decay with detuning in the P supply as a result of reduced belowground carbohydrate allocation [136]. In a word, carbohydrate depletion in roots deters P nutrition in young forest trees [136]. It has been found that P acquisition, plant internal P cycling and soil processes collectively affect P nutrition in beech ecosystems [137]. Due to the lack of an experimental method suitable for natural forests, P absorption of trees in the field is difficult to be verified. An alternatively novel approach, ^18^O-labelled ^31^P-phosphate (^31^P^18^O_4_^3−^), could be used to measure P acquisition in the field. Measurements of ^18^O-P_i_ absorption show the rapid metabolism of Pi in the roots of trees [137].

Recently, a study has identified some tree miRNAs from three sRNA libraries which are constructed from *Populus tomentosa* Carr. subjected to sufficient or Pi depletion condition or to the restoration of a sufficient Pi level after Pi deficiency [138]. Four miRNAs, pto-miR167, pto-miR394, pto-miR171, and pto-miR857, can not only be responsive to low N but also be implicated in the response to low Pi [138,139]. To be specific, pto-miR394 and pto-miR857 were massively restrained and induced under Pi deficiency conditions, respectively [138]. In addition, the abundance of pto-miR6135k, pto-miR6441, pto-miR167, pto-miR171, and pto-miR827 altered greatly under Pi starvation [138]. Profiling of miRNAs and their targets has uncovered a possible miRNAs-mediated interaction network to respond to Pi deprivation in *Betula luminifera* [140]. *PHOSPHOLIPASE D DELTA* (*PLDD*), a member of the *PHOSPHOLIPASE D* (*PLD*) family, was impeded in an emblematic way for PLD family proteins and it was found to be a cleavage of miR399c [140]. It is presumed that *PLDD* acts as a negative regulator in *B. luminifera* under LP conditions [140]. Accordingly, inducing the isomiRs like miR399c to decrease the *PLDD* abundance in LP-treated roots of *B. luminifera* might enhance the tolerance to Pi starvation [140]. miR397 and miR398 were upregulated in both roots and shoots of Pi starved *B. luminifera*. It is speculated that miR397 participates in the regulation of *laccase* genes in shoots, and increased miR397 expression might lead to the reduced mRNA levels, taking part in the lignification and thickening of cell wall under -Pi conditions in *B*. *luminifera* [140]. In *B*. *luminifera*, miR169c was downmodulated in Pi-treated roots and *NFYA1*, as well as NFYA10 which act as the targets, were vastly induced [140]. So far, other responses of miR169-NFYAs to Pi starvation are still unclear. miR395b was induced and *ATP SULFURYLASE 1* (*APS1*), the target of miR395b, was downregulated in -Pi treated roots of *B. luminifera* [140].

## 7. Significance of Study on the Utilization Efficiency of P in Forest Trees

Many studies about the P seeking and P tolerance mechanisms have been carried out worldwide in recent years. However, the current review reports are mostly limited to crops, and the low P stress of forest trees review reports on adaptability are rare. Forest trees are the material basis of forestry production and the largest ecosystem on land, accounting for about 82% of terrestrial biomass, widely distributed on the earth with rich biodiversity, accounting for more than 50% of terrestrial biodiversity [127,141]. Not only can they provide a large number of forestry products for human beings, but also play an influential role in maintaining the steady state of the biosphere effect [127,142,143]. The best beautification function and ecological function of trees count on the normal growth and development of trees in their growing environment. For example, the survival and growth status of urban greening trees are not only determined by the genetic characteristics of trees but also hampered by environmental factors. Drought, extreme temperature (high temperature, low temperature), flood, and saline are common environments curbing the normal growth and development of trees.

According to the results of the Ninth National Continuous Forest Resources, in inventory from 1973 to 2018, China’s plantation area is 79.54 million hm^2^, with a total volume of 3387.5996 million m^3^, the highest in the world [144]. Among the 31 provinces (autonomous regions and municipalities directly under the Central Government), Guangxi has the largest plantation area of 7,335,300 hm^2^, accounting for 9.22% of the country’s total area, and Tibet is 78,400 hm^2^, which is the smallest. Actually, two-thirds of the country’s suitable forest land is distributed in the arid and semi-arid areas of the five Northwest Provinces and Inner Mongolia Autonomous Region, and 17.99% of the suitable forest land is planned on steep, abrupt, and dangerous slopes above 25°, with poor site conditions, great afforestation difficulty, and high afforestation cost. P is an important factor restricting the growth and development of trees, so how to improve the utilization efficiency of P is particularly profound.

## 8. Efficient Strategies to Fasten the P Cycle in Natural System

Environmental agents greatly affect the P acquisition of plants [145,146,147]. It is believed that soil pH is the host variable, owing to its significant influences on lots of chemical reactions including toxic waste, essential plant nutrients, and phytotoxic elements [147]. pH could influence the P solubility, which determines mobility and bioavailability. With respect to the maximum P availability of pH is controversial, many studies have been focused on the problem. On the one hand, some researchers pretend that the maximum phosphate availability occurs in the pH range 6 to 7 [146,147]. On the other hand, others insist that P absorption by plants and desorption by soil occur with “a much lower pH optimum” [147]. Since there are several different P retention mechanisms, it is wrong to assume that only one P retention mechanism is occurring, except perhaps for pure mineral systems [147]. In fact, the core of the issue is that there are three distinct influences of pH. For example, when the pH is declined from 6 to 4, the phosphate absorption rate rises, the extent desorbed from soil adds, and the amount absorbed by soil increases as well [147]. The availability is improved by the first two influences but reduced by the third effect [147]. To complicate matters further, the pH of the root surface may be different from that of the massive soil and be substantially affected by the form of N supply and plant species [147]. Plentiful studies have indicated that soil edaphic properties and plant species diversity could be affected by different land-use types [148,149]. In the meanwhile, it has been reported that soil P constrains biodiversity across European grasslands [148]. Researchers have demonstrated that independent of the environmental impacts of N pollution, plant species richness was consistently negatively related to soil P, based on a dataset covering 501 grassland plots throughout Europe [148,150]. In addition, a similar phenomenon is found in both the Australian savanna and in Northeast American pastures [151,152]. Based on the complicated soil chemical processes that control how soil pH affects P availability to plants and the influence of land use/cover on P availability, it is of great necessity for us to fasten the P cycle in a natural system.

Based on the above, we could draw the following conclusions. A large amount of transcription factors and transporters have been characterized, that seem to modulate PSR, either positively or negatively. The positive regulators existing in P sensing and signaling, absorption, transportation, storage, metabolism, and other pathways are *AtPHR1*, *AtPHL1*, *AtPHL2*, *AtPHL3*, *AtMYB2*, *AtWRKY45*, *AtWRKY75*, *AtPHT1*, *AtIPS1/2, OsPHR2, OsWRKY21, OsWRKY108, OsWRKY74, OsPTF1, Os**PHT1, OsIPS1/2*, miR399, miR827, etc., which not only help to enhance using efficiency of Pi but also make a difference to growth characters including LRWC, Pn, Fv/Fm, transpiration rate (*Tr*), proline content, superoxide dismutase (SOD) activity, etc. *AtMYB62*, *AtWRKY6*, *AtWRKY42*, *AtbHLH32*, *AtZAT6*, *AtSPX1*, *AtSPX2*, *OsIPS1/2*, miR395 and miR398 operate as repressors in such processes. Several potential and feasible tactics are proposed to fasten the P cycle. First of all, it is possible that knock out negative regulators or overexpress the positive regulators to achieve the goal of improving the efficiency of Pi. Secondly, genetic improvement of P by plants could be carried out according to different site conditions. It is well known that acidic soil, alkaline soil, and neutral soil which are classified depending on pH have distinct levels of P. Acidic soil and alkaline soil contain lower Pi compared to neutral soil. On the one hand, we can advance the perceptual sensitivity of P to improve P-acquisition efficiency in areas with low Pi content; on the other hand, enhance the P-use efficiency to drive growth in areas with comparatively high Pi content. In addition, drought resistance and water-saving characteristics could be increased by reinforcing the P-use efficiency in arid and semi-arid lands. Meanwhile, improving P-acquisition efficiency, photosynthetic P-use efficiency, and P-remobilization efficiency are favorable ways to fasten the P cycle in plants. These three tactics are as follows. Firstly, plant P-acquisition efficiency could be improved via root foraging tactics and P-mining tactics which are associated with root features, including architectural, morphological, physiological, and symbiotic features [10]. For example, there are more adventitious roots and lateral branches, thinner and longer roots, more root hairs, and more organic anions and phosphatases are released, as well as AMF, ectomycorrhizal fungi, and P-solubilizing bacteria existing in roots and so on [10]. Studies have found that phosphorus-acquisition strategies of different plants have considerable genotype differences such as common bean (*Phaseolus vulgaris*), maize (*Zea mays*), chickpea (*Cicer arietinum*), and poplar (*Populus simonii*) [10]. It will be a fantastic research direction to ameliorate P-acquisition efficiency that uses alternative methods which are easy to measure to pay more attention to the genotypic change in plants in roots-related microbial ways [10]. Agronomic tactics are also a good way to improve P acquisition in different soils including acid soil, calcareous soil, and neutral soil. For instance, intercropping high P-mining plants and optimizing P-application rates in the proper stage to meet plant P requirements are common agronomic methods to enhance plants P-acquisition efficiency and improve plants development, growth, yield, and quality subsequently [10]. We can integrate one of the formers with the latter.

It is well known that in the plant canopies, not all leaves operate with complete photosynthetic capacity, and they only operate under light conditions. Most of the upright leaves maintain a faster photosynthetic rate and higher photosynthetic phosphorus-use efficiency (PPUE) because of the large effective photosynthetic surface area, while the PPUE of broad-leaved plants is lower on account of the negligible infiltration of light into the lower leaf layer. Thus, how to achieve the rapid retransfer of P in the lower leaf layers before the senescence of broad-leaved plants is crucial [10]. Some time ago, an interesting phenomenon was discovered by researchers, that is, P signal transduction is able to regulate the tilt of rice leaves, which shows that P can conduct the plant canopy, thereby realizing effective photosynthesis [143]. As the natural changes of P metabolism and signal transduction in plants are closely related to the level of PPUE, it is of practical significance for us to search the natural changes of P metabolism and signal transduction in plants, and to ensure and test varieties with effective PPUE [10]. Under the conditions of fast photosynthesis and high PPUE, how to achieve the balance between distribution of P for different leaf cell types and P pools has become a key issue for plants to operate normally [145,153]. To keep the balance between them, plants have evolved an ingenious mechanism, which has diverse performance in different dimensions. For example, P has a preferential allocation to photosynthetic cells in monocots or eudicots evolving in insufficient P places when the PPUE is high at the level of leaf [154,155]. In the meanwhile, at the level of the cell, sulfolipids, and galactolipids replace phospholipids, which work when ribosomal RNA contents are low [156,157]. There is a positive correlation between PPUE and the ratio of metabolic P to lipid P [157]. It is vital for us to comprehend the operating mechanism to accomplish the objective of cultivating plant varieties or varieties that have high PPUE. The P distribution model might not only help to reduce the application rate of P fertilizers, but also contribute to increasing plant production.

In addition, the process of P remobilization within the plant becomes particularly important when the supply of soil P is inadequate. There are a lot of cellular processes that promote P remobilization, such as the decomposition of the plasma membrane, the substitution of membrane phospholipids by lipids without P, the decomposition of smaller P esters, and the hydrolysis of RNA due to incremental ribonuclease activity, which all prepare for the remobilization of P, for example, the transport of P from aging organs to young and active organs which function as one of the P sources for plants such as rice, maize and wheat [158,159,160,161]. In summary, having much knowledge of the cellular processes that facilitate P remobilization, the underlying impacts of environmental agents on the processes, and the differences in the effectiveness of exogenous P among different varieties is critical to enhance P-remobilization efficiency and then formulate appropriate plant enhancement plans.

## 9. Perspective

P in the form of Pi is an important nutrient for growth, development, and reproduction in plants, however, Pi is one of the least available essential nutrients owing to its low available concentrations and insolubility [40,47]. In response to the availability of Pi in soil, plants have evolved a wide range of responses, including the reduction of internal Pi usage and activation of external Pi acquisition and recycling, which are mediated by a multidimensional signaling network of hormones, sugars, non-coding RNAs, other nutrients and other regulatory factors that play a role at different levels of regulation [48,105]. In addition, plant growth could be greatly affected by biotic and abiotic stresses, but the relation between Pi starvation and other environmental stress signaling pathways is still opaque. As a matter of fact, the existence of such a network system could ensure that plants survive the absence of Pi, which is usually happening in nature. The organization of these signal pathways and the coordination of time and space are intriguing. Nowadays, we have a good understanding of which genes are concerned with P uptake and how plants absorb P from the soil, but the molecular mechanism of how P is transported in plant tissues and how Pi signals are transmitted between cells after entering plant cells are unclear. Future attention should be paid to examining the interactions between components of signaling pathways as well as finding a new signaling mechanism. Furthermore, it is necessary for us to develop the systems to monitor signals and the input and outcome of the Pi signaling pathways. Understanding how different signal paths are integrated to provide a diverse and dynamic response is the main problem facing them. With the progress of advanced technology, the face of the Pi signaling network will be more comprehensive. Using genomic techniques (e.g., GWAS) to excavate topflight natural allelic variation and genome editing techniques (e.g., CRISPR/Cas9) to create elite allelic variation will be potential ways in phosphate efficiency breeding in the near future. All in all, this knowledge will be conducive to future plant breeding, with the aim of enhancing the efficiency of P acquisition and utilization.

## Figures and Tables

**Figure 1 ijms-23-02353-f001:**
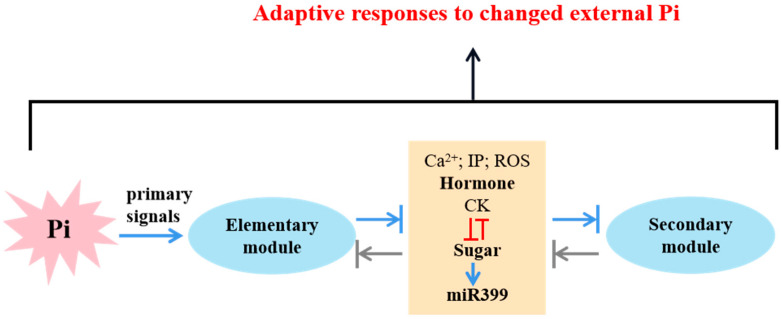
A summary of proposed phosphate signaling cascades. The diversity of pathways is activated by the single input of an elementary signal, resulting in a lot of responses. In addition, the potential secondary signal components are shown in the model as well, acting as a center to transmit the elementary signal to the secondary module.

**Figure 2 ijms-23-02353-f002:**
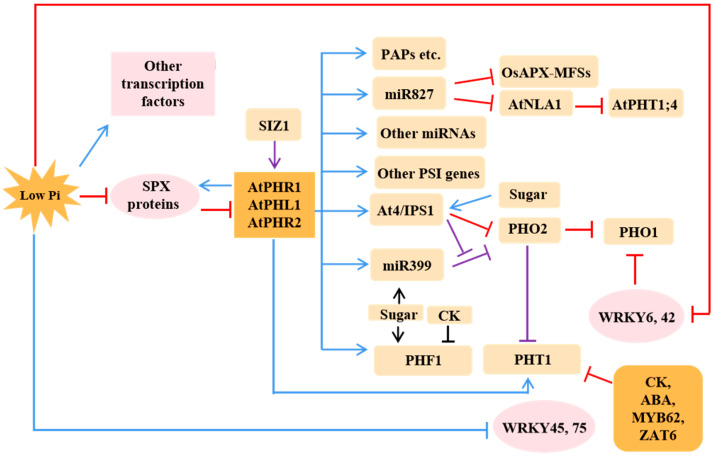
Molecular regulation pathway of PSRs. The colored lines show the mode of action at the levels of transcription and post transcription. Black lines represent still undefined modes, blue lines represent the transcriptional regulation and purple lines represent the posttranscriptional regulation. Arrows denote positive effects, whereas lines ending with a short bar indicate negative effects.

**Figure 3 ijms-23-02353-f003:**
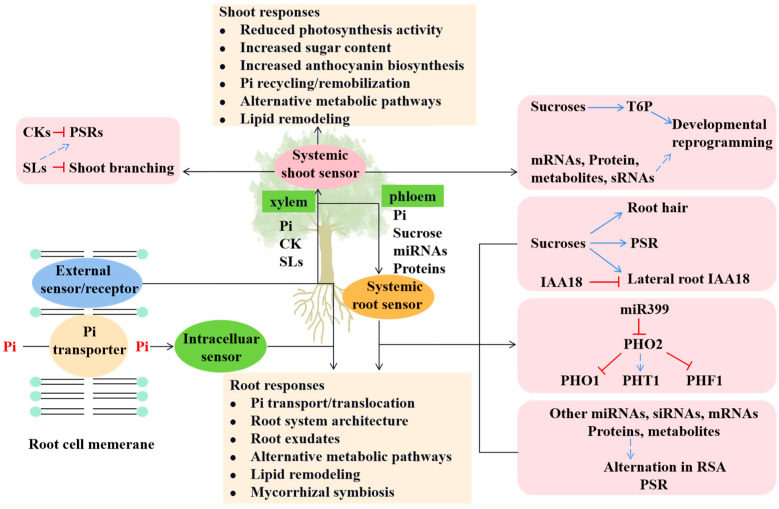
The model of local and systemic phosphate sensing and related signaling pathways in plants. Phosphate could be sensed extracellular or intracellular to induce local signaling pathways in roots and the synthesis of systemic signals. The responses reflected on shoots and roots to Pi deficiency are listed separately.

**Table 1 ijms-23-02353-t001:** Transcriptional factors involved in Pi signaling.

Type of Factor	Name	Locus	Reference
MYB Family	*PHR1*	At4g28610	[49]
	PHL1	At5g29000	[53]
	PHL2	At3g24120	[53,54]
	PHL3	At4g13640	[51,53]
	PHL4	At2g20400	[55]
	MYB62	At1g68320	[56]
	MYB2	At2g47190	[57]
WRKY Family	WRKY6	At1g62300	[58,59]
	WRKY42	At4g04450	[60]
	WRKY45	At3g01970	[61]
	WRKY75	At5g13080	[62]
BHLH Family	BHLH32	AT3g25710	[50]
Other families	ZAT6	AT5g04340	[61]

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
