# Peer review of "The Genetic Basis of Phosphorus Utilization Efficiency in Plants Provide New Insight into Woody Perennial Plants Improvement"

_ijms, 2022, doi:10.3390/ijms23042353_

Round 1
Reviewer 1 Report
Your manuscript entitled: "The genetic basis of phosphorus utilization efficiency in plants provide new insight into woody perennial plants improvement", proposes some of sufficient possible sustainable management strategies to fasten the P cycle in modern agriculture. Also, your manuscript adequately reviews recent progresses in mechanism underlying P availability-responsive P acquisition, the transportation, storage and metabolism of P and the genetic basis of P utilization. To increase the manuscript’s visibility and the readers’ interest, it would be better for this review effort to include a paragraph for the effects of environmental factors on P acquisition.
Comment 1: In section 8. Efficient strategies to fasten the P cycle in natural system
I would suggest adding a paragraph on soil chemical processes that control how soil pH affects phosphorus availability to plants. Many studies are available (eg. Penn, C. J., & Camberato, J. J. (2019). Agriculture, 9(6), 120.; Barrow, N.J. Plant Soil 2017, 410, 401–410.; Lindsay, W. L., & Stephenson, H. F. (1959). Soil Science Society of America Journal, 23(1), 12-18.). Also, many studies have reported that land use/cover affects phosphorus availability to plants (eg. Triantafyllidis, et al., (2020). Effect of land-use types on edaphic properties and plant species diversity in Mediterranean agroecosystem.; Ceulemans, T., et al.( 2014). Soil phosphorus constrains biodiversity across European grasslands); (obviously, they are not exhaustive of the literature).
Author Response
Our response: We thank the Reviewer 1 for this constructive suggestion. The paragraph of soil chemical processes that control how soil pH affects phosphorus availability to plants has been added in section 8 according to your suggestion. In addition, relevant studies that land use/cover could affect phosphorus availability to plants also have been added in current manuscript.

Reviewer 2 Report
This paper is a review centered on the genetic and molecular basis of P utilization in plants.
In my opinion the paper is well done and acceptable for publication, after a few minor revision, and specifically my concern is about:
1. Figure resolution and quality should be improved and standardized.
2. The references must be carefully checked: some references lack italic characters in botanical names, reference 143 has some words highlighted, there are capital letters instead of lowercase letters, etc.
Author Response
Our response: We thank the Reviewer 2 for this constructive suggestion. I am sorry for these mistakes. They have been corrected in current manuscript revision.
